# Influence of Alternate Hot and Cold Thermal Stimulation in Cortical Excitability in Healthy Adults: An *f*MRI Study

**DOI:** 10.3390/jcm9010018

**Published:** 2019-12-19

**Authors:** Sharon Chia-Ju Chen, Jau-Hong Lin, Jui-Sheng Hsu, Chiu-Ming Shih, Jui-Jen Lai, Miao-Ju Hsu

**Affiliations:** 1Department of Medical Imaging and Radiological Sciences, Kaohsiung Medical University, Kaohsiung 80708, Taiwan; sharchen@kmu.edu.tw; 2Department of Medical Research, Kaohsiung Medical University Hospital, Kaohsiung 80708, Taiwan; jhlin@kmu.edu.tw; 3Department of Physical Therapy, Kaohsiung Medical University, 100 Shih-Chuan 1st Road, Kaohsiung 80708, Taiwan; 4Department of Medical Imaging, Kaohsiung Medical University Hospital, Kaohsiung 80708, Taiwan; e3124@ms16.hinet.net (J.-S.H.); shihchiuming3927@gmail.com (C.-M.S.); 890223@kmuh.org.tw (J.-J.L.)

**Keywords:** alternative thermal stimulation, cortical excitability, crossover experimental design, finger-tapping task, functional MRI

## Abstract

Stroke rehabilitation using alternate hot and cold thermal stimulation (*alt*TS) has been reported to improve motor function in hemiplegia; however, the influence of brain excitability induced by *alt*TS remains unclear. This study examined cortical activation induced by *alt*TS in healthy adults, focusing on motor-related areas. This involved a repeated crossover experimental design with two temperature settings (innocuous *alt*TS with alternate heat-pain and cold-pain thermal and noxious *alt*TS with alternate heat and cold thermal) testing both arms (left side and right side). Thirty-one healthy, right-handed participants received four episodes of *alt*TS on four separate days. Functional magnetic resonance imaging scans were performed both before and after each intervention to determine whether *alt*TS intervention affects cortical excitability, while participants performed a finger-tapping task during scanning. The findings revealed greater response intensity of cortical excitability in participants who received noxious *alt*TS in the primary motor cortex, supplementary motor cortex, and somatosensory cortex than in those who received innocuous *alt*TS. Moreover, there was more motor-related excitability in the contra-lateral brain when heat was applied to the dominant arm, and more sensory-associated excitability in the contra-lateral brain when heat was applied to the nondominant arm. The findings highlight the effect of heat on cortical excitability and provide insights into the application of *alt*TS in stroke rehabilitation.

## 1. Introduction

Advances that utilize neuroplasticity in physical rehabilitation are emerging, particularly for patients with hemiplegic stroke [1]. Many rehabilitative techniques based on remodeling neural circuits and enhancing brain plasticity have been developed to facilitate recovery of motor functions [2,3,4].

Thermal stimulation (TS) is a potential approach to inducing cortical reorganization along somatosensory pathways. The response intensity and regions of activated corticomotor excitability induced by TS are linked strongly to the temperature of TS [5,6,7]. The neurologic effect of thermally induced cortical excitability, the so-called neurologic pain signature, has been reported to act in individually separate ways in the innocuous and/or noxious thermal pathways [8]. Innocuous TS stimulates thermal receptors and activates the primary somatosensory cortex (S1, known as the postcentral cortex), the secondary somatosensory cortex (S2, known as the superior–inferior parietal cortex in part), the thalamus, and insula [6,9]. Meanwhile, noxious TS stimulates nociceptors and effectively activates lateral and medial pain systems that involve motor association areas, including the supplementary motor cortex (SMA) and the anterior cingulate cortex (ACC) [7,10,11].

As the brain activities evoked by TS overlap each other in the motor-related cortex, TS is considered to have potential in helping in recovery of the motor functions of extremities affected by stroke [12,13]. Given the corticomotor excitability induced by TS, clinicians and researchers have developed a new therapeutic approach in stroke rehabilitation that involves alternate TS (*alt*TS) by applying heat and cold alternately to the paraplegic/paretic limbs in a single TS intervention protocol. Chen et al. [14] were the first to report the use of hot and cold packs to provide alternate cycles of heat-pain (temperature >45 °C) and cold-pain stimulation (temperature <5°C) to the paretic extremities of patients with acute stroke. They reported significant motor and functional recovery of paretic extremities. A similar finding was made by Wu et al. [12], who custom-made equipment to provide an *alt*TS intervention at constant preset temperatures for patients with subacute stroke. Furthermore, our colleagues Hsu et al. [13] compared the *alt*TS effect at varying thermal combinations of either noxious temperature with alternate heat-pain and cold-pain thermals (*n-alt*TS) or innocuous temperature with alternate heat and cold thermals (*in-alt*TS) in patients with stroke. Their findings indicated that the *alt*TS group with a noxious temperature combination showed significant improvements in the outcome measures, such as the Fugl–Meyer scale and the Barthel index, whereas the innocuous temperature group exhibited no improvement according to the findings of motor and functional measurements [13]. Similarly, Lin et al. [15] reported that, compared with *in-alt*TS, *n-alt*TS performs better on the Fugl–Meyer scale after a two-month *alt*TS intervention (three times a week at 30 min per time). Furthermore, recently, Chen et al. applied *n-alt*TS combined with neuromuscular electrical stimulation and seemed to help upper-extremity recovery after stroke [16]. Although the evidences of previous studies demonstrated that *alt*TS intervention can improve motor and functional recovery of the paretic limbs in patients with acute and subacute stroke, the relationship between the appropriate combination of thermal treatment and promotion of function is inconclusive in terms of the optimal benefits for patients with stroke.

Although clinical measures provide evidence of the efficacy of *alt*TS in stroke rehabilitation [15,16], understanding of the underlying regulation associated with the hot-and-cold (*alt*TS) thermal processing for corticomotor excitability is very limited. Only one study by transcranial magnetic stimulation (TMS) showed that a 30 min noxious *alt*TS increased motor-evoked potentials and mapping size of the affected hemisphere in stroke patients in comparison with innocuous *alt*TS [17]. Even though TMS assessment can detect changes in neuronal activity, the obtainable observable actions found are limited to a local area response of the brain. The overall functional alterations within and between brain regions can hardly be determined. To increase understanding of *alt*TS-induced corticomotor excitability by fMRI (functional magnetic resonance imaging), we conducted this cross-over experiment using two manipulated factors and acquired a series of *f*MRI readings before and after the *alt*TS interventions to investigate the *alt*TS effect. The manipulated two thermal parameters were (1) by varying alternate hot-and-cold thermal combinations (*in-alt*TS and *n-alt*TS thermal settings) and (2) by applying *alt*TS on both sides of the body for right-handed participants, who had left-hemisphere-dominant brains.

## 2. Materials and Methods

### 2.1. Participant Preparation and General Protocol

Thirty-one healthy right-handed participants were recruited (17 men, age 21.7 ± 2.1 years). The inclusion criteria included (1) no history of cognitive deficit and either neurological or psychological disease; (2) no skin injuries, burns, or fresh scars on the upper extremities; and (3) no musculoskeletal or neuropathic disease affecting the upper extremities. We asked all participants to abstain from using caffeine, tea, alcohol, and nicotine for at least one day before the day of the experiment. All participants reported that they are genuinely right-handed in everyday tasks. After having the nature of the study explained, the participants provided written consent, which has been approved by the Institutional Review Board of Kaohsiung Medical University Chung-Ho Memorial Hospital (protocol code: KMUH-IRB-20110369). All experimentation and analysis in this study was conducted in accordance with the Declaration of Helsinki.

We used a randomized and repeated crossover experimental design to reduce the effect of confounding covariates and achieve statistical efficiency through full conditional repeats. Subjects were required to receive four repeated measurements in four different weeks for a long enough washout time to thermal experience. The sequence of four measurements was assigned in a random order according to two factors: the side affected (left arm and right arm) and the temperature combination of the *alt*TS (*in-alt*TS and *n-alt*TS). The *in-alt*TS was the temperature alterations of 42 °C and 17 °C, while the *n-alt*TS was the temperature alterations of 51 °C and 4°C, respectively [12,13]. We tested the randomization of the experimental sequence using a Friedman’s test (Equation (1), *p*-value = 0.76). On each day of the experiment, participants received an *alt*TS intervention based on the assigned experimental procedure, that is, right arm applied with *in*TS or left arm applied with i*n*TS and so on, and we conducted *f*MRI scans both before and after the *alt*TS. Participants were asked to perform a finger-tapping task during the *f*MRI scanning.
(1)$$\chi_{3}^{2}=1.18$$

### 2.2. altTS Equipment and Intervention

We used a custom-made thermal generator that included two individual hot and cold system modules (FISTEK, Model-B300, B401L, Diagnostic & Research Instruments Co, Ltd., Taiwan). Each module had a flexible therapeutic pad (38 × 55 cm, Gaymar TP22E, Gaymar Industries, Inc., USA) to transmit the assigned temperature to the affected arm via circulating water within individual plastic tubes. The temperature was predefined on the generator’s console panel and varied between 2 °C and 3 °C at the therapeutic pad. The *alt*TS modules are depicted in Figure 1a.

The *alt*TS intervention (in which heat and then cold were applied twice each) was conducted while the participants were seated comfortably and their arms were resting on pillows. The *alt*TS protocol was as described in previous studies [13,14]. Figure 1b provides a brief introduction to the *alt*TS protocol, which consisted of two cycles of *alt*TS intervention with alternating heating and cooling blocks. During the heating block, the hot pad was wrapped around the participant’s affected forearm for 15 s, followed by a 15 s break (no TS application), with 10 repetitions. After a break interval of one minute, the cold pad was wrapped around the same affected forearm for 30 s, followed by a 15 s break (no TS application), also with 10 repetitions. After an inter-cycle break lasting three minutes, the same heating and cooling blocks were repeated. The total duration of the *alt*TS protocol was approximately 30 min. During the *alt*TS intervention, participants were allowed to withdraw their arms if they became unable to bear the temperature.

We used two *alt*TS temperature combinations to examine the contrasting influence of temperature on cortical excitability. One was the innocuous temperature combination (hot: 42 °C, and cold: 17 °C) and the other was the noxious temperature combination (hot–pain: 51 °C, and cold–pain: 4 °C) [12,13]. The assigned *alt*TS protocol (innocuous *alt*TS and noxious *alt*TS) was applied to the participant’s affected arm (left or right) on each day of the experiment.

### 2.3. MRI Scanning Protocol

We used a 3-Tesla GE Signa HDX MRI machine (Milwaukee, WI, USA) for the *f*MRI study. We acquired functional images using the T2*-weighted gradient echo planer imaging sequence, which measured the blood oxygen level dependence (BOLD) signal. We acquired a time series of 165 repetitions (330 s) with the following scan parameters: slice thickness = 4 mm, in-plane resolution = 3.4 × 3.4 mm^2^, TR/TE/*θ* = 2000 ms/30 ms/90°, matrix size = 64 × 64, and 28 axial slices of the whole brain per repetition. We removed the first ten volumes for signal stability, and then the rest data (155 volumes) was entered into data analysis. We acquired another set of anatomic T1 images using the three-dimensional spoiled gradient-recalled acquisition in steady state (256 × 256 × 124 voxels for the whole brain scan, with a voxel resolution of 1 × 1 × 1 mm). We used this anatomic imaging as the template on which we overlapped the selected functional activations.

We performed *f*MRI scans both before and after the *alt*TS interventions to measure the effect of *alt*TS on cortical excitability, using the following sequence: anatomic T_1_, *f*MRI, *alt*TS intervention, *f*MRI, and anatomic T_1_. Participants were asked to perform a functional task of finger tapping during the *f*MRI scanning.

### 2.4. A Functional Task of Finger Tapping

The finger tapping task is used commonly in *f*MRI studies to assess brain function in terms of motor performance [18,19]. We modeled the induced hemodynamic response in a function of the difference of two gamma-variate functions (Γ), as shown in Figure 2a, [20] and modulated it according to the cognitive task. We used a conventional block design that alternated five blocks of active period with six blocks of rest period, with 30 s intervals, as shown in Figure 2b. We asked each participant to follow instructions projected onto a pair of goggles (CinemaVision, Resonance Technology, Inc., Canada) and to perform finger tapping at the rate of one finger per second during the active motor period, with continued finger movements for 30 s, followed by a 30 s rest period without finger movement. During the finger-tapping period, the thumb had to come in contact with each finger.

### 2.5. fMRI Preprocessing and Data Analysis

We processed the acquired 3D + time *f*MRI data using statistical parametric mapping (SPM, version 8.0) analysis, implemented in the MATLAB (Math Works) environment. SPM8 was developed by Friston’s Lab (Wellcome Trust Centre of London University, UK) for analyzing *f*MRI data; its results are presented as a statistical parametric map with statistical values. The data analysis toolkit in MATLAB codes is available for free download. The data analysis procedure included data preprocessing and a following statistical test. Preprocessing steps included (1) inter-slice timing correction to the middle slice of a volume for signal coherence, (2) realignment of the motion artifact to the first volume in the whole scanning period, (3) co-registration/normalization of spatial transformation to the standardized head domain merged from 512 participants’ imaging results (the Montreal Neurological Institute template MNI, called the MNI template), and (4) spatial smoothing with a Gaussian kernel of 4 mm full to half maximum to elevate the signal-to-noise ratio of functional responses. We later resampled the preprocessed data into 61 (*x*-axis) × 73 (*y*-axis) × 61 (*z*-axis) × 155 (volumes) at a voxel resolution of 3 × 3 × 3 mm.

We have used a model-based method to analyze functional alteration during the finger-tapping task. As shown in Figure 2c, we regarded a responsive curve of the hemodynamic response function as the convolution of a canonical response kernel (a default gamma-variant function in Figure 2a) and the experimental design protocol, as shown in Figure 2b. Based on the functional connection between similar functional regions, we calculated the correlation intensity according to the task-related dependence across voxels in the brain, as shown in Figure 2d. Then, we presented the intensity map of correlation with the correlation coefficient after Pearson correlation calculation between hemodynamic response function and the time course of each brain voxel. The obtained correlation intensity was later transformed into statistical value after Z-transformation ($z=\frac{r_{i}-\bar{r}}{\sigma_{r}}$), as shown in Figure 2e.

### 2.6. Statistical Analysis

As for the global functional alteration due to thermal stimulation, we used statistical methods to explore the thermal influence on motor function after *alt*TS intervention based on the statistical model of factorial design. In all statistical tests, the preprocessed data (the responsive intensity presented as z-value, shown in Figure 2e) was inputted into the statistical model as the dependent variable. The significance of the model was tested between independent variable (functional response) and dependent variables (the interest factors). The statistically parametrized outcome for brain voxels was next extracted as the significantly activated region according to the statistical value above the significance level of *p*-value 0.05, which is null hypothesized as H_0_: equivalence between compared samples. The extracted voxels were tested for suitability in a cluster by an alpha probability simulation (AlphaSim method), which is used to ascertain whether the extracted voxels within a cluster were prevented from the probability of random noise within a threshold cluster by a given *t*-value threshold (*p* < 0.05). Here, we conducted a one-factor repeated ANOVA (*p* < 0.05) to test the thermal effect between two interventions of *n*-*alt*TS and *in*-*alt*TS, using *pre*-*alt*TS data as the controlled covariate. The *pre*-*alt*TS data as the covariate variable was applied to decrease the variation in the regression model caused by heterogeneity of baseline between compared independent variables. Accordingly, we presented the quantified information of activated clusters with the number of voxels within the cluster, the maximum *F*-value (*F*_max_) within a cluster, and the coordination of *F*_max_.

In addition, in our analysis of two motor-related regions, the precentral cortex and the postcentral cortex, we conducted a one-factor repeated-measure ANOVA adjusted by a covariate with *pre*-TS data to test the responsive difference of functional alteration between the two interventions of *n*-*alt*TS and *in*-*alt*TS for each arm. We not only considered the thermal effect between different temperature settings, but also examined the interactive relationship between arms treated and thermal treatments applied by using two-factor repeated-measure ANOVA adjusted by *pre*-TS data as the covariate (*p* < 0.05). Here, we have investigated the main effects of the arm side and the thermal settings and their interactions.

## 3. Results

### 3.1. Participant’s Report

All 31 participants completed the study and reported no adverse effects, such as burning or blistering, during *alt*TS.

After the *alt*TS interventions, participants were asked to report their sensations with both thermal settings of *alt*TS, for example, “Did you want to withdraw your arm during the TS? If so, which temperature TS combination makes you want to draw back your arm?” Seven participants (four males and three females) were unable to tolerate the *n-alt*TS setting and withdrew their arms during the intervention. Of these seven subjects, two females and two males could not stand the hot–pain thermal treatment applied to the right arm (arm*R*). One female and two males could not stand either the hot–pain or the cold–pain thermal treatment applied to the left arm (arm*L*). In our exploratory observation, their signal performances did not go beyond twice the standard deviation around the group mean. Therefore, to keep the statistical power of comparison between groups, we adopted all samples for the inference in the statistical analysis.

### 3.2. Thermal-Induced Cortical Excitability

We examined the thermal effect (*in-alt*TS vs. *n-alt*TS) on different arm sides using one-way repeated ANOVA with the controller covariate, data of *pre-alt*TS. After threshold of statistical significance at the level of *p*-value of 0.05 after correction, the influenced brain regions are shown in Figure 3. The activation maps of subjects for each particular condition was presented in supplement material (Appendix A).

As seen in Figure 3, we found that, when the thermal treatment was applied to arm*L*, the bilateral activations included the ACC, the superior frontal cortex, the superior parietal cortex, and the middle-superior temporal cortex, and the unilateral activations included the left angular cortex, the right middle frontal cortex, the left hippocampus, the right insula, the left lingual gyrus, the left middle occipital cortex, the left parahippocampus, the right postcentral cortex, the right precentral cortex, the left precuneus, the right putamen, the right operate rolandic gyrus, and the right supramarginal gyrus. Conversely, thermal treatment of arm*R* caused activations that were located mainly in the left hemisphere, including the left middle-superior frontal cortex, the left precentral cortex, the left postcentral cortex, and bilateral SMAs. To summarize these overall activations, application of *alt*TS on arm*L* mainly influenced brain regions of the bilateral basal ganglia, whereas application of *alt*TS on arm*R* induced activation more in bilateral SMAs and the left postcentral cortex.

Table 1 shows the quantitative information of the activation regions in respect of brain hemisphere, activated voxels, maximum statistical value (*F*-max) of the cluster, and the coordination of the *F*-max value for each region activated. The brain activation upon thermal stimulation indicated that there may have been different effects depending on the arm affected. In Table 1, *n-alt*TS applied to arm*L* induced greater activation (1216 voxels over 23 brain regions above statistical significance (*p* < 0.05)) compared with application on arm*R* (623 activated voxels over 11 brain regions). The greatly influenced regions (*F*_1,60_ >15) were located consistently on the side of the brain contralateral to the arm treated. For arm*L*, *n-alt*TS induced high levels of activation in the middle frontal cortex, the superior parietal cortex, the putamen, and the precentral cortex, whereas, for arm*R*, the high-level activation was in the precentral cortex and the postcentral cortex. Table 1 highlights the distribution of influenced regions for each arm treated. For arm*L*, *n-alt*TS induced more activation distributed more broadly than was the case for arm*R*.

### 3.3. Functional Excitability in the Precentral Cortex, Postcentral Cortex, and SMA

To understand the motor excitability due to thermal stimulation in detail, we have selected two motor-related areas (i.e., the precentral cortex and the postcentral cortex) to demonstrate the modulation of functional plasticity by varying the thermal settings of *alt*TS (*in-alt*TS and *n-alt*TS). In Figure 4a–d, the thermal effect, defined as the activated intensity after correction of baseline with *pre-alt*TS data (shown with a dash line), shows that a higher effect was generated consistently by *in-alt*TS than by *n-alt*TS. Statistical examination of the two thermal settings was conducted using a repeated-measure ANOVA, and it revealed a consistently significant difference beneath the level of *p*-value 0.05.

Accordingly, we conducted further statistical analysis for the precentral cortex and the postcentral cortex using the two-factor repeated-measure ANOVA adjusted by a covariate, *pre*-TS data, to test the relationship between the two factors of affected arm sides and thermal settings (*F*_1,58_ = 43.88, *p*-value < 0.01). In Table 2, the main arm effect showed a significant impact of thermal perception on the affected arm side (both regions, *p*-value < 0.05), and the thermal main effect showed a more significant impact on the brain responses by the applied temperature (both regions, *p*-value < 0.01). Moreover, the influence of interaction was insignificant for both regions (*p*-value > 0.05), suggesting that the thermal effect on cortical excitability might not be impacted by the arm side affected.

## 4. Discussion

Studies of clinical outcome measures have shown that *alt*TS helps motor and functional recovery in patients with stroke [13,14,17]. However, neuroimaging-based evidence concerning the cortical reorganization associated with *alt*TS is limited. This study is the first human study using fMRI to examine how *alt*TS affects motor cortical excitability through varying heat-and-cold thermal combinations. The main results supported our hypotheses and indicated that *n-alt*TS induced a prominent functional alteration of motor-related activity that is dependent on the arm side affected, in comparison with innocuous *alt*TS. When we applied *n-alt*TS to arm*L* (the nondominant arm), the significantly influenced regions were distributed mainly in the bilateral basal ganglia and right insula, the right precentral cortex (mainly the primary motor cortex, M1), the right postcentral cortex (the primary somatosensory cortex, S1), and the bilateral superior parietal cortex (the primary somatosensory cortex, S2). When we applied *n*-*alt*TS to arm*R* (the dominant arm), the significantly influenced regions were distributed mainly in motor-related areas, including bilateral SMAs, the left precentral cortex (mainly the primary motor cortex, M1), and the left postcentral cortex (the primary somatosensory cortex, S1).

In this study, we varied the temperature for the combination of hot-and-cold thermal within the *alt*TS intervention and examined the distribution of motor excitability in the ipsilateral and contralateral brain of the affected arm. If we contrasted the temperature to a greater extent on the dominant arm, there was a positive portion of activity in motor-related brain regions, such as M1, S1, and SMA, whereas, with the nondominant arm, there was greater activation in the sensory-related regions, such as the ACC, the insula, and S2. TS induced thermal-associated neuronal activity in the motor areas through the thermal sensation pathway. This effect of TS on healthy individuals and animal models demonstrated that noxious TS is involved in functional promotion via the lateral and medial pain systems through the thermal sensation pathway [9]. Davis et al. [5] applied cold and hot thermal stimulation separately in different sessions (temperature modulated along with thermal intensity from either cool to cold pain or warm to hot pain) for 40–60 sec on the right hand for normal subjects and found that noxious TS promoted activity mostly in motor-related areas (M1 and SMA), the lateral and medial thalamus, the anterior insula, and contralateral S2. Moreover, this thermal-induced aftereffect was proportional to a thermal intensity up to 46 °C, and the sensory-discriminative processing of pain was confined to the SMA [5]. These findings supported the notion that TS can drive functional plasticity in the brain, and our study findings further support the idea that the extent of functional plasticity is also dependent on thermal temperature and the arm side affected.

Compared with *in-alt*TS, our study showed that *n-alt*TS contributed to increased brain excitability by greater functional alteration in areas, such as M1, SMA, S1, and S2, which echoed to the expression for the pathway of the neurologic pain signature [8]. Some studies have reported that cortical excitability, such as in M1, S1, and S2, can be promoted through somatosensory stimulation heads to specific task-related activations [21,22,23]. In the schema of thermal transmission, S1 receives direct input from the affected hand and has direct anatomic connection with M1, the premotor region, and S2 [22,23]. These connections modulate neuronal activity in the M1 and associated areas, providing a likely anatomic substrate for the thermal effects reported in animal studies [24,25]. Furthermore, by creating a large temperature contrast between hot and cold treatment, *n-alt*TS can increase the alteration of activation of the motor cortical function by modulating GABAergic neurotransmission and altering long-term potentiation-like processes [26,27]. Recent studies have reported explicitly that the capsaicin receptor, including subfamily M and V of the transient receptor potential cation channel (TRP), is involved in thermal processing through both somatosensory neurons and autonomic neurons [28]. Thermal stimulation has been shown to activate ion channels over the entire thermal range, from extreme heat, to painful heat, to warmth, to coolness, to painful cold with TRPV2, TRPV1, TRPM3, TRPV3, TRPV4, TRPM2, TRPM8, and TRPA1 [28,29,30]. These observations provide a possible explanation why thermal sensation can alter motor function directly. Manipulation of temperature between hot and cold thermals would promote the responsive intensity of the neuronal activity via the somatosensory pathway, leading to M1 excitability.

Notably, we have also found that the thermal effect on cortical excitability is expressed differently depending on the arm side affected, reflecting that expression of functional lateralization is involved in the thermal intervention. Application of the thermal treatment on the dominant arm produced functional variation not only in the contralateral motor areas (i.e., M1, S1, and SMA) but also in the contralateral anterior frontal cortex and ipsilateral SMA. Conversely, this thermal effect for the nondominant arm drove functional responses at the contralateral motor areas (i.e., M1 and S1) and also at the ipsilateral insula, the bilateral S2, and the bilateral temporal cortex. Moreover, we tested the arm effect and the interactive effect of arm × thermal (arm side and thermal setting) in two main motor-related areas (M1 and S1, as shown in Figure 4) and found that their interactive effect was insignificant. This suggested that, to understand the thermal effect on cortical excitability, one must consider the arm side affected and the temperature individually. In other words, functional lateralization in a brain hemisphere has its own alteration process based on thermal variation. As mentioned in other studies, brain asymmetry applies not only to anatomical structures but also to the difference in the functional specialty of both hemispheres [31]. Handedness is also a factor in functional asymmetry between hemispheres in individuals [24,32]. Cabinio et al. [33] found that M1 and S2 are lateralized prominently in the contralateral hemisphere in right-handed study participants when the subjects are either observing or moving their right hand. However, under thermal stimulation, we found that the arm’s sensation of heat does affect functional specialty. In this study, we have focused mainly on examining the effect of different thermal temperatures on cortical excitability and have overlooked the fact that functional alteration is affected by the arm side treated. Our findings have shown that thermal intervention induced more motor activity in the contralateral brain of the dominant arm, but it induced more sensory activity in the contralateral brain of the nondominant arm for right handedness. However, to differentiate whether the thermal effect would carry out the excitability efficacy due to the brain’s functional lateralization, further investigation in which the thermal treatment is applied to the affected arm of participants with varying handedness would be worth conducting.

Given that thermal pain is a multidimensional integration of sensation and perception, its prolonged effects on the functional plasticity of motor functions [6] and the heterogeneous pain-related outcomes with individual’s pain experience [8] are complex. Thermal pain involves significant modulations of functional processing, such as cognitive evaluation of features of painful stimulation, affect, attention, and motor control [34]. The prolonged effect of adaptation to thermal pain is spread mainly through two core regions: the thalamus and the somatosensory cortex [5,6]. The main pathways of thermal pain travel via the lateral and the medial thalamus. Accordingly, the affiliated pathway of cognitive function during TS can contribute to functional performance. Distinguishing the relationship between the pain intensity of TS and cognitive performance improves understanding of the neural substrate in motivational and emotional responses to pain [21,35,36,37]. The pain intensity of TS at varying temperatures under either innocuous or noxious thermal conditions mediates pain processing in the thalamus, the insula, the ACC, S1, S2, SMA, and the premotor areas; it also affects the performance of sensation, motor control, and attention to some extent [38]. Thermal pain–induced brain regulation has been confirmed by current academic understanding of neurologically functional pathways and electrophysiological nociception [8,39] passing through the spino-thalamic tract [30]. Taken this advantage, our approach might be helpful in models of neuropathic pain following acute and chronic spinal cord injuries (e.g., degenerative cervical myelopathy) in the application of neurotechnological field [40]. In our study, activation of the thalamus, which is considered to be the center of thermal sensing, has not been presented in the results because it might be expressed in the prompt effect of TS and not in the prolonged effect, given that our experimental design lasted 30 min.

Not only did we use a crossover experimental design and conduct motor tasks to demonstrate explicitly the elevated statistical power of the analysis of motor-associated areas, with the significance criterion set at *p* < 0.05, but also we used a large stimulation area with a thermal pad to increase the skin’s sensation of temperature and controlled the treatment delivery at a stable temperature using circulating water [13]. These experimental preparations allowed us to observe motor excitability. However, we must acknowledge some limitations in our study. We detected the thermal effect by comparing the functional alteration after thermal stimulation. However, the instant measure of functional alteration caused by thermal variation during hot and cold duration was not involved during TS intervention. As such, the dynamic functional alteration might be omitted. In addition, we examined neuroplasticity by using a single session of *alt*TS. Longitudinal follow-up after *alt*TS intervention is required to examine functional regulation and adaptation. Future studies should also consider examining whether our results can be applied either to the lower limbs or to stroke populations.

To summarize, our findings showed that *n-alt*TS was involved in greater functional excitability of motor-related brain activities than was *in-alt*TS. Furthermore, the thermal influence on cortical excitability was biased by the arm side affected. Application of *alt*TS to the dominant arm contributed more to motor-related function, whereas application to the nondominant arm contributed more to sensory response than to motor excitability. These results provide insights into the underlying exploration of the effect of *alt*TS on neuroplasticity under alternate heat-and-cold treatment and illustrate the potential application of *alt*TS for stroke neurorehabilitation.

## Figures and Tables

**Figure 1 jcm-09-00018-f001:**
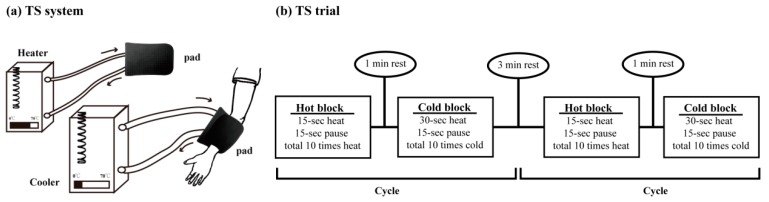
The alternate thermal stimulation (*alt*TS) protocol. (**a**) The *alt*TS device includes two independent heat and cold generators to deliver thermal treatment at a specific temperature through individual circulating tubes. (**b**) The *alt*TS intervention procedure includes two repeats, in which heat and cold are applied alternately at specific temperatures.

**Figure 2 jcm-09-00018-f002:**
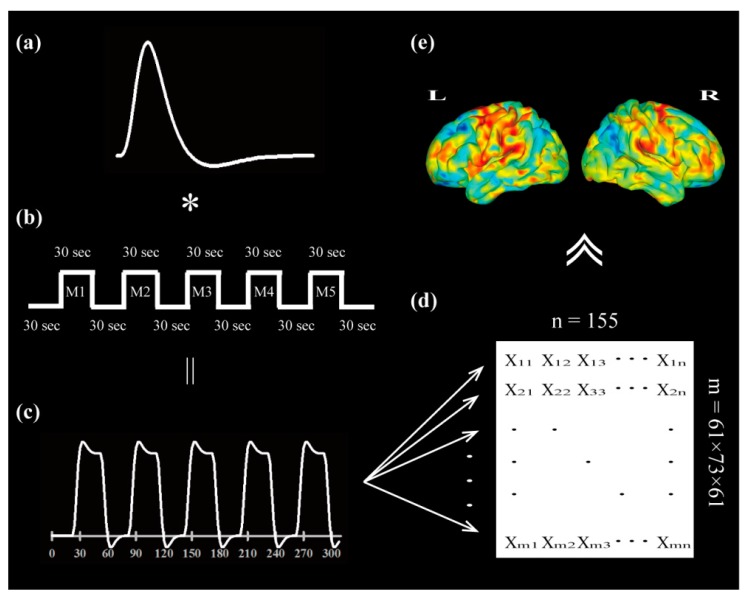
Procedure of analysis of imaging data. All imaging data acquired were extracted for the task-related signal through detection of signal matching between the responsive BOLD (blood oxygen level dependence) signal and the modeled hemodynamic response function. (**a**) The experimental design protocol was convolved using a default canonical hemodynamic response function. (**b**) A blocked experimental design with five repeated finger-tapping periods was used. Accordingly, (**c**) a modeled hemodynamic response function was generated to screen whole spatial voxels across time. (**d**) Task-related response intensity was calculated by Pearson correlation calculation between hemodynamic response function and the time response course of voxel across whole brain. (**e**) The activation map represented with statistical value (z-value) was shown. The color range from minimum negative z-value to maximum positive z-value was presented from cold color to warm color.

**Figure 3 jcm-09-00018-f003:**
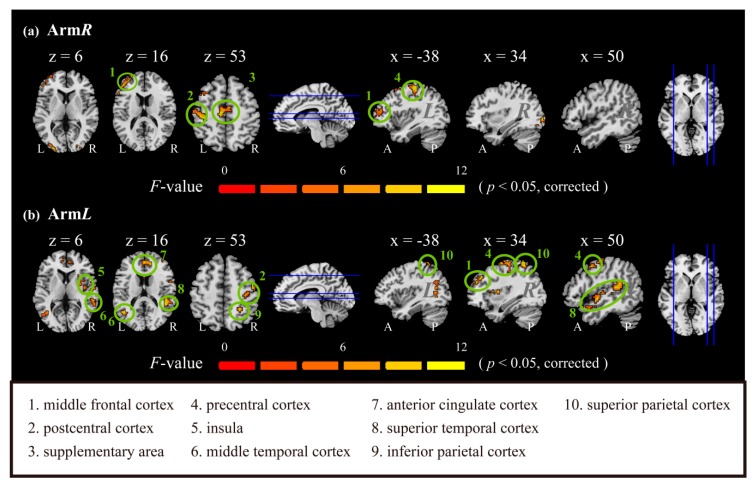
Thermal effect for two arms after statistical examination. The effects of *alt*TS applied to two arms (**a**) arm*L* (left arm) and (**b**) arm*R* (right arm) were tested using one-way repeated ANOVA analysis, with a covariate of pre-TS data. After multiple comparisons, the highlighted brain regions indicating significant activation difference between *in-alt*TS and *n-alt*TS were singled out above the statistically significant level of *p*-value 0.05. The color bar presents the value of statistical F-value. *p*-value was corrected by the Alphasim method as the correction of the multiple comparisons, which was described in the Statistical Analysis Session of Materials and Methods.

**Figure 4 jcm-09-00018-f004:**
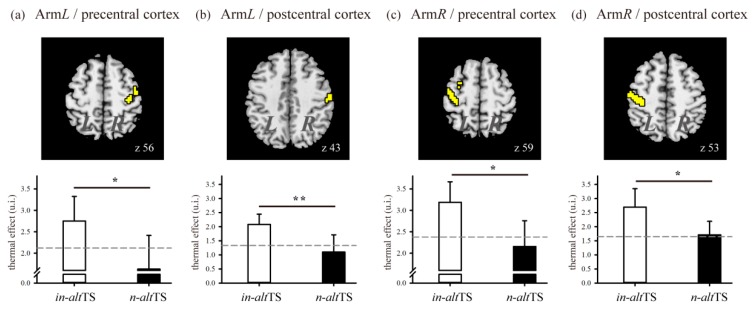
Thermal effect in two major motor-related areas. The thermal effect for arm*L* shown in (**a**) the right precentral cortex and in (**b**) the right postcentral cortex. The effect for arm*R* shown in (**c**) the left precentral cortex and in (**d**) left postcentral cortex. The difference between *in-*altTS and *n-alt*-TS was studied using a repeated-measure ANOVA with a covariate of the data of *pre*-TS. The adjusted base line of *pre*-TS data is presented as a gray dash line. (*: *p*-value < 0.05; **: *p*-value < 0.01). Abbreviation: arm*L*, left arm; arm*R*, right arm; *in-alt*TS, innocuous *alt*TS; *n-alt*TS, noxious *alt*TS.

**Table 1 jcm-09-00018-t001:** Influenced regions of thermal treatment (in-altTS vs. n-altTS) for each affected arm. One-way repeated ANOVA with pre-TS data as a covariate was used to study the difference between *in-alt*TS and *n-alt*TS.

Arm*L*	Side	Voxels	F-Max ^a^	X	Y	Z	Arm*R*	Side	Voxels	F-Max ^a^	X	Y	Z
AngularC	L	38	7.92	−42	−66	30	InfTri-FrontalC	L	84	9.35	−51	27	30
Ant-CingulumC	L	38	10.49	0	48	12	Mid-FrontalC	L	136	10.57	−33	51	15
	R	57	12.72	15	42	15	Sup-FrontalC	L	15	5.79	−27	63	6
Mid-FrontalC	R	101	17.91	30	42	27	Inf-OccipitalC	L	16	7.02	−24	−99	−9
Sup-FrontalC	L	18	8.31	−12	51	24		R	37	9.01	30	−84	−12
	R	14	12.03	30	−6	66	Mid-OccipitalC	L	43	12.54	−27	−96	6
Hippocampus	L	25	9.75	−24	−30	−6		R	23	12.43	33	−93	0
Insula	R	44	11.16	33	9	9	PostcentralC	L	114	19.82	−39	−27	57
LingualG	L	19	8.07	−21	−54	−6	PrecentralC	L	62	24.05	−36	−27	60
Mid-OccipitalC	L	21	8.45	−36	−63	21	SuppMotor-Area	L	52	14.68	−3	−15	60
ParaHippocampus	L	28	11.02	−33	−42	−6		R	41	14.14	3	−12	60
Sup-ParietalC	L	69	10.15	−33	−51	60							
	R	58	15.64	30	−51	66							
PostcentralC	R	81	11.85	57	−18	45							
PrecentralC	R	147	16.39	48	−9	54							
Precuneus	L	13	6.74	−12	−66	66							
Putamen	R	29	15.11	30	9	9							
Oper-Rolandic	R	16	7.35	57	6	9							
SupraMarginal	R	24	9.84	45	−36	24							
Mid-TemporalC	L	96	12.74	−45	−63	15							
	R	69	11.07	48	−45	15							
Sup-TemporalC	L	11	9.33	−54	−3	−12							
	R	200	15.16	60	−9	−9							

These regions were selected based on a statistical criterion (*p* < 0.05 after correction). ^a^: maximum *F*-value within the cluster. Abbreviation: Ant-CingulateC, anterior cingulate cortex; Mid-FrontalC, middle frontal cortex; Sup-FrontalC, superior frontal cortex; Mid-OccipitalC, middle occipital cortex; Sup-ParietalC, superior parietal cortex; Mid-TemporalC, middle temporal cortex; Sup-TemporalC, superior temporal cortex; InfTri-FrontalC, inferior-triangular frontal cortex; Inf-OccipitalC, inferior occipital cortex; SuppMotor Area, supplementary motor area; *in-alt*TS, innocuous *alt*TS; *n-alt*TS, noxious *alt*TS.

**Table 2 jcm-09-00018-t002:** Comparison of the thermal effect on two applied arms on selected motor-related areas. Two motor-related areas, the precentral cortex and the postcentral cortex, were used to evaluate the thermal effect. A two-factor ANOVA with a covariate of *pre*-TS data was used to study the relationship between the factors applied arms (arm*L* and arm*R*) and applied *alt*TS (*in-alt*TS and *n-alt*TS).

Regions	ArmL		ArmR		Statistics		
	*in-alt*TS	*n-alt*TS	*in-alt*TS	*n-alt*TS	Arm	Thermal	Arm × Thermal
Precentral cortex	2.75 (0.59)	1.63 (0.78)	3.19 (0.44)	2.16 (0.60)	4.21 *	17.85 **	0.02
Postcentral cortex	2.08 (0.36)	1.10 (0.61)	2.70 (0.65)	1.71 (0.48)	6.41 *	15.90 **	0.01

Cells show the mean statistical value (standard deviation) over participants, and statistical values represent the functional alteration after *alt*TS. ^*^ Significance at *p* < 0.05; ^**^ significance at *p* < 0.01. Abbreviations: *n-alt*TS, noxious *alt*TS; *in-alt*TS, innocuous *alt*TS.

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
