# Peer review of "Influence of Alternate Hot and Cold Thermal Stimulation in Cortical Excitability in Healthy Adults: An fMRI Study"

_jcm, 2019, doi:10.3390/jcm9010018_

Round 1
Reviewer 1 Report
The article presents interesting studies on the impact of thermal stimulation on the motor functions of healthy people based on functional magnetic resonance imaging studies.
Detailed fMRI analysis of the sensorimotor cortex has shown that cortical activations depend on the type of thermal stimulation and the location of application of stimulation on both arms (left vs right).
The authors showed good knowledge of literature on the subject of research in both the introduction and discussion. However I have some remarks: 1) term “ dominant brain” should be specified (Introduction), 2) the use of two temperature setting ( innocuous altTS vs noxious altTS is not clearly presented in Materials and Methods.
On the other hand statistical analysis based on one and/or two factor repeated measure ANOVA should be underlined.
the information obtained can be useful in planning of the physical therapy in stroke patients. However, these findings require validation in clinical medicine.
Author Response
Journal: Journal of Clinical Medicine (jcm-659366)
Title: Influence of alternate hot and cold thermal stimulation in cortical
excitability in healthy adults: an fMRI study
Subject: Response to Reviewers
Reviewer #1 (Remarks to the Author):
1. The article presents interesting studies on the impact of thermal stimulation on the
motor functions of healthy people based on functional magnetic resonance imaging
studies.
=> Thank you for the comment.
2 Detailed fMRI analysis of the sensorimotor cortex has shown that cortical activations
depend on the type of thermal stimulation and the location of application of stimulation
on both arms (left vs right).
=> Thank you for the comment.
3. The authors showed good knowledge of literature on the subject of research in both
the introduction and discussion. However I have some remarks: 1) term “dominant
brain” should be specified (Introduction), 2) the use of two temperature setting
(innocuous altTS vs noxious altTS is not clearly presented in Materials and Methods.
=> Thank you for the suggestion.
1). The term “dominant brain” is clarified in the introduction as follows.
Abstract, Page 1, “there was more motor-related excitability in the contra-lateral
brain when heat was applied to the dominant arm and more sensory-associated
excitability in the contra-lateral brain when heat was applied to the nondominant
arm.”
Discussion, Page 19, LN 16-18, “In this study, we varied the temperature for the
combination of hot-and-cold thermal within the altTS intervention and
examined the distribution of motor excitability in the ipsilateral and
contralateral brain of the affected arm.”
Discussion, Page 21, “Our findings have shown that thermal intervention
induced more motor activity in the contralateral brain of dominant arm, but it
induced more sensory activity in the contralateral brain of non-dominant arm
for right handedness.”
2). As suggested, the following statements are added to the Method. Page 5, LN 6-8:
“The in-altTS was the temperature alterations of 42°C and 17°C, while the
n-altTS was the temperature alterations of 51°C, and 4°C, respectively (Wu
2
et al., 2010; Hsu et al., 2013).”
4. On the other hand statistical analysis based on one and/or two factor repeated
measure ANOVA should be underlined.
=> Thank you for the suggestion. As suggested, the statistical analysis used has been
implemented in Table 1 and Figure 3.
5. The information obtained can be useful in planning of the physical therapy in stroke
patients. However, these findings require validation in clinical medicine.
=> Thank you for the comment. We agree with the reviewer. In the discussion section,
relevant information is stated on Page 22, LN 9-12 as follows. ”We examined
neuroplasticity by using a single session of altTS. Longitudinal follow-up after
altTS intervention is required to examine functional regulation and adaptation.
Future studies should also consider examining whether our results can be
applied either to the lower limbs or to stroke populations.”

Reviewer 2 Report
In the manuscript “Influence of alternate hot and cold thermal stimulation in cortical excitability in healthy adults: an fMRI study”, Chen et al., investigated the differences between brain activation after the alternate hot and cold thermal stimulation (altTS) in the innocuous and noxious levels. This study contributes to the understanding of the underlying mechanism of the therapeutic effect of altTS. Several concerns need to be addressed.
For each fMRI session, 160 volumes (320s) were acquired. However, the block design was 5 active and 6 rest, 30s of each, which should be 330s in total. Please clarify this discrepancy.Page 7, Line 15, subjects were asked to “perform self-paced finger tapping”, while in Line 17 “the thumb had to come in contact with each finger at the rate of one finger per second”. Please confirm whether the pace of finger tapping was fixed or not.
Was detrending, band-pass filtering and motion parameters regression performed in fMRI preprocessing?
Figure 2e demonstrated activation map of task-related response but there is no figure caption describing the details about the map, e.g., what is the color scale, is the map a pre-altTS condition or post-altTS condition...
The process of generating the activation map and extracting significantly activated regions was described in the method, while results were not demonstrated. Showing activation maps of all four conditions (controlled for baseline) would help readers to appreciate the difference between in-altTS and n-altTS.
Authors mentioned about extracting significantly activated regions. But it is not clear what was done with these regions. Regions were extracted based on p < 0.05. Is it the corrected p-value? Please provide more details.
In the “statistical analysis” section, for the voxel-wise analysis, was thermal effect tested on the coefficient value which was described in the “fMRI preprocessing and data analysis” section? For the analysis of the precentral cortex and the postcentral cortex, it is mentioned in the result section that thermal effect was defined as the activated intensity after correction of baseline with pre-altTS data. Please describe the details of how activated intensity was calculated and how it was baseline-corrected in the method.
“Participants were asked to report their sensations with both thermal settings of altTS”. What question or assessment was used to evaluate their sensation? Please include the report in the result if possible. Seven participants withdrew during intervention. Were they included in the data analysis? If so, were their data different from others?
Figure 3, based on the description in the first paragraph of page 11, the highlighted brain regions indicate significant activation difference between in-altTS and n-altTS instead of “significant impact of in-altTS and n-altTS” as stated in the figure caption. Please also add information about the color scale in the figure caption (warmer color indicates higher activation after n-altTS?). Please confirm whether the value of the color scale was t or F.
In the manuscript, authors refer to the statistically significant regions as “activated regions”. However, these regions are not regions that are “activated” during finger tapping (it is unknown because results not shown). They are regions that showed significantly different activation between in-altTS and n-altTS. The term “activated regions” is confusing.
Page 12, line 14, authors compared the F value from repeated ANOVA. However, large F does not mean the effect is large, and vice versa. To determine whether the n-altTS vs. in-altTS difference is larger in specific regions, authors need to compare the difference directly between regions (e.g., percentage of changes in region A vs. region B).
Page 15, “In Table 2, the main arm effect showed a significant impact of thermal perception on the affected arm side (both regions, p-value < 0.05), and the thermal main effect showed a more significant impact on the applied temperature (both regions, p-value < 0.01).” Please rephrase “the thermal main effect showed a more significant impact on the applied temperature” because applied temperature is the independent variable so it will not be affected by the thermal main effect.
Page 19, Line 3, Coghill et al., 1999 does not support the statement “the sensory-discriminative processing 2 of pain was confined to the SMA”.
In discussion section, it should be stated clearly that the observed alteration is based on comparison with innocuous TS, instead of the effect of thermal treatment which may be taken as post-altTS vs. pre-altTS.
This study reported an arm side difference of the thermal effect. Is there previous study showing this difference in terms of treatment outcomes?
Typo:
Page 4, Line 4: “cannot hardly be determined” should be “can hardly be determined”
Figure 4 caption “The thermal effect for armR shown in (a) the right precentral cortex and in (b) the right postcentral cortex. The effect for armL shown in (c) the left precentral cortex and in (d) left postcentral cortex.” “armR” and “armL” are flipped.
Page 20, Line 2, “different” should be “differently”
Author Response
Journal: Journal of Clinical Medicine (jcm-659366)Title: Influence of alternate hot and cold thermal stimulation in cortical excitability in healthy adults: an fMRI study
Subject: Response to Reviewers
Reviewer #2 (Remarks to the Author):
In the manuscript “Influence of alternate hot and cold thermal stimulation in cortical excitability in healthy adults: an fMRI study”, Chen et al., investigated the differences between brain activation after the alternate hot and cold thermal stimulation (altTS) in the innocuous and noxious levels. This study contributes to the understanding of the underlying mechanism of the therapeutic effect of altTS. Several concerns need to be addressed.
=> Thank you for the comment.
For each fMRI session, 160 volumes (320s) were acquired. However, the block design was 5 active and 6 rest, 30s of each, which should be 330s in total. Please clarify this discrepancy.
=> Thank you for the suggestion. We have rephrased the sentence in Page 7 Line 2-6 as “We acquired a time series of 165 repetitions (330 s) with the following scan parameters: slice thickness = 4 mm, in-plane resolution = 3.4 × 3.4 mm2, TR/TE/θ = 2000 ms/30 ms/90°, matrix size = 64 × 64, and 28 axial slices of the whole brain per repetition. We removed the first five volumes for signal stability and then the rest data (160 volumes) was entered into data analysis.”
Page 7, Line 15, subjects were asked to “perform self-paced finger tapping”, while in Line 17 “the thumb had to come in contact with each finger at the rate of one finger per second”. Please confirm whether the pace of finger tapping was fixed or not.
=> Thank you for the comment. The subject was asked to perform finger tapping as a fixed rate, with no cues. The relevant statement was revised as follows. “We asked each participant to follow instructions projected onto a pair of goggles (CinemaVision, Resonance Technology, Inc., Canada) and to perform finger tapping at the rate of one finger per second during the active motor period, with continued finger movements for 30 s followed by a 30-s rest period without finger movement. During the finger tapping period, the thumb had to come in contact with each finger.”
Was detrending, band-pass filtering and motion parameters regression performed in fMRI preprocessing?
=> Thank you for your comment. Generally speaking, the process of “detrending, band-pass filtering” was used for the data analysis of resting state fMRI study. In this study, we adapted model-based method based on the pattern of experimental design. The associated description was addressed in Page 9 Line 14-23 as “We have used a model-based method to analyze functional alteration during the finger tapping task. As shown in Figure 2c, we regarded a responsive curve of the hemodynamic response function as the convolution of a canonical response kernel (a default gamma variant function in Figure 2a) and the experimental design protocol, as shown in Figure 2b. Based on the functional connection between similar functional regions, we calculated the correlation intensity according to the task-related dependence across voxels in the brain, as shown in Figure 2d. Then, we presented the intensity map of correlation with the correlation coefficient after Pearson correlation calculation, as shown in Figure 2e. Subsequently, we extracted the significantly activated regions based on the statistical value above the significance level of p-value 0.05.”.
(1) Figure 2e demonstrated activation map of task-related response but there is no figure caption describing the details about the map, e.g., what is the color scale, is the map a pre-altTS condition or post-altTS condition...
(2) The process of generating the activation map and extracting significantly activated regions was described in the method, while results were not demonstrated.
(3) Showing activation maps of all four conditions (controlled for baseline) would help readers to appreciate the difference between in-altTS and n-altTS.
=> Thank you for your comment.
(1) Figure 2e was described in Page 8 in the legend of Figure 2 as “The activation map represented with statistical value (z-value) was shown. The color range from minimum negative z-value to maximum positive z-value was presented from cold color to warm color.”.
(2) The process of generating the activation map and extracting significantly activated regions was modified in the “fMRI preprocessing and data analysis” as follows.
Page 9: “We have used a model-based method to analyze functional alteration during the finger tapping task. As shown in Figure 2c, we regarded a responsive curve of the hemodynamic response function as the convolution of a canonical response kernel (a default gamma variant function in Figure 2a) and the experimental design protocol, as shown in Figure 2b. Based on the functional connection between similar functional regions, we calculated the correlation intensity according to the task-related dependence across voxels in the brain, as shown in Figure 2d. Then, we presented the intensity map of correlation with the correlation coefficient after Pearson correlation calculation between hemodynamic response function and the time course of each brain voxel. The obtained correlation intensity was later transformed into statistical value after Z-transformation ( ), as shown in Figure 2e”
In the “Statistical analysis” in Page 10:
“we used statistical methods to explore the thermal influence on motor function after altTS intervention based on the statistical model of factorial design. In all statistical tests, the preprocessed data (the responsive intensity presented as z-value shown in Figure 2e) was inputted into statistical model as the dependent variable. The significance of model was tested between independent variable (functional response) and dependent variables (the interest factors). The statistically parametrized outcome for brain voxels was next extracted as the significantly activated region according to the statistical value above the significance level of p-value 0.05 which is null hypothesized as H0: equivalence between compared samples. The extracted voxels were tested its suitability in a cluster by an alpha probability simulation (AlphaSim method) which is used to ascertain whether the extracted voxels within a cluster were prevented from the probability of random noise within a threshold cluster by a given t-value threshold (P < 0.05). Here, we conducted a one-factor repeated ANOVA (P < 0.05) to test the thermal effect between two interventions of n-altTS and in-altTS, using pre-altTS data as the controlled covariate. The pre-altTS data as the covariate variable was applied to decrease the variation in the regression model caused by heterogeneity of baseline between compared independent variables.”
(3) The difference between in-altTS and n-altTS has been examined and shown the activation maps of all four conditions (controlled for baseline) in the following and in “Supplemental Material”.
Figure. The activation maps of subjects for each particular condition (2 factors: arm and TS) before and after thermal stimulation. In the (a) armL-inTS condition (upper-left map), the activated areas included the ventral part of frontal lobe, posterior cingulate cortex, left precentral cortex, left postcentral cortex and middle cingulate gyrus. In the (b) armR-inTS condition, the aforementioned motor-related areas, insula and left inferior parietal lobe were activated. In the (c) armL-nTS and (d) armR-nTS conditions, a bilateral activation in the precentral cortex, postcentral cortex (S1), supplementary motor cortex and superior parietal lobe (S2) was shown. The statistical significance threshold was set at p < 0.01 for multiple comparisons.
Authors mentioned about extracting significantly activated regions. But it is not clear what was done with these regions. Regions were extracted based on p < 0.05. Is it the corrected p-value? Please provide more details.
=> Thank you for your comment. We applied AlphaSim method to do the multiple comparison for p-value correction. The associated description about the extraction of activated region and the correction of p-value was described in Page 10 as “we used statistical methods to explore the thermal influence on motor function after altTS intervention based on the statistical model of factorial design. In all statistical tests, the preprocessed data (the responsive intensity presented as z-value shown in Figure 2e) was inputted into statistical model as the dependent variable. The significance of model was tested between independent variable (functional response) and dependent variables (the interest factors). The statistically parametrized outcome for brain voxels was next extracted as the significantly activated region according to the statistical value above the significance level of p-value 0.05 which is null hypothesized as H0: equivalence between compared samples. The extracted voxels were tested its suitability in a cluster by an alpha probability simulation (AlphaSim method) which is used to ascertain whether the extracted voxels within a cluster were prevented from the probability of random noise within a threshold cluster by a given t-value threshold (P < 0.05). Here, we conducted a one-factor repeated ANOVA (P < 0.05) to test the thermal effect between two interventions of n-altTS and in-altTS, using pre-altTS data as the controlled covariate. The pre-altTS data as the covariate variable was applied to decrease the variation in the regression model caused by heterogeneity of baseline between compared independent variables.”
In the “statistical analysis” section, for the voxel-wise analysis, was thermal effect tested on the coefficient value which was described in the “fMRI preprocessing and data analysis” section? For the analysis of the precentral cortex and the postcentral cortex, it is mentioned in the result section that thermal effect was defined as the activated intensity after correction of baseline with pre-altTS data. Please describe the details of how activated intensity was calculated and how it was baseline-corrected in the method.
=>Thank you for your comment. (1) Data preprocessing was described in the “fMRI preprocessing and data analysis” in Page 9 as “We have used a model-based method to analyze functional alteration during the finger tapping task. As shown in Figure 2c, we regarded a responsive curve of the hemodynamic response function as the convolution of a canonical response kernel (a default gamma variant function in Figure 2a) and the experimental design protocol, as shown in Figure 2b. Based on the functional connection between similar functional regions, we calculated the correlation intensity according to the task-related dependence across voxels in the brain, as shown in Figure 2d. Then, we presented the intensity map of correlation with the correlation coefficient after Pearson correlation calculation between hemodynamic response function and the time course of each brain voxel. The obtained correlation intensity was later transformed into statistical value after Z-transformation ( ), as shown in Figure 2e.”and in the “Statistical analysis” in Page 10 as “we used statistical methods to explore the thermal influence on motor function after altTS intervention based on the statistical model of factorial design. In all statistical tests, the preprocessed data (the responsive intensity presented as z-value shown in Figure 2e) was inputted into statistical model as the dependent variable. The significance of model was tested between independent variable (functional response) and dependent variables (the interest factors). The statistically parametrized outcome for brain voxels was next extracted as the significantly activated region according to the statistical value above the significance level of p-value 0.05 which is null hypothesized as H0: equivalence between compared samples. The extracted voxels were tested its suitability in a cluster by an alpha probability simulation (AlphaSim method) which is used to ascertain whether the extracted voxels within a cluster were prevented from the probability of random noise within a threshold cluster by a given t-value threshold (P < 0.05). Here, we conducted a one-factor repeated ANOVA (P < 0.05) to test the thermal effect between two interventions of n-altTS and in-altTS, using pre-altTS data as the controlled covariate. The pre-altTS data as the covariate variable was applied to decrease the variation in the regression model caused by heterogeneity of baseline between compared independent variables.”
“Participants were asked to report their sensations with both thermal settings of altTS”. What question or assessment was used to evaluate their sensation? Please include the report in the result if possible. Seven participants withdrew during intervention. Were they included in the data analysis? If so, were their data different from others?
=> (1) The subject was asked to respond to general questions, such as “do you feel any discomfort, or pain?” The associated description has described in Page 11 Line 7-9 as “After the altTS interventions, participants were asked to report their sensations with both thermal settings of altTS, for example “Did you want to withdraw your arm during the TS? If so, which temperature TS combination makes you want to draw back your arm?”.”.
(2) We have added sentences for the description in dealing with the unexpected phenomenon of data in Page 11 Line 13-16 as “In our exploratory observation, their signal performances didn’t go beyond twice standard deviation around group mean. Therefore, to keep the statistical power of comparison between groups, we adopted all samples for the inference in the statistical analysis.”.
Figure 3, based on the description in the first paragraph of page 11, the highlighted brain regions indicate significant activation difference between in-altTS and n-altTS instead of “significant impact of in-altTS and n-altTS” as stated in the figure caption. Please also add information about the color scale in the figure caption (warmer color indicates higher activation after n-altTS?). Please confirm whether the value of the color scale was t or F.
=> Thank you for the opinion. We modified the description of legend in Figure 3 in Page 12 as “After multiple comparisons, the highlighted brain regions indicating significant activation difference between in-altTS and n-altTS were singled out above the statistically significant level of p-value 0.05. The color bar presented the value of statistical F-value. P-value was corrected by the Alphasim method as the correction of the multiple comparisons which was described in the Statistical Analysis Session of Materials and Methods.”.
In the manuscript, authors refer to the statistically significant regions as “activated regions”. However, these regions are not regions that are “activated” during finger tapping (it is unknown because results not shown). They are regions that showed significantly different activation between in-altTS and n-altTS. The term “activated regions” is confusing.
=>Thank you for the comment. We changed “activated regions” to“highlighted regions” or “influenced regions”.
Page 12, line 14, authors compared the F value from repeated ANOVA. However, large F does not mean the effect is large, and vice versa. To determine whether the n-altTS vs. in-altTS difference is larger in specific regions, authors need to compare the difference directly between regions (e.g., percentage of changes in region A vs. region B).
=> Thank you for your comment. We agree that large F value from repeated ANOVA analysis does not mean the effect is large, and vice versa. To determine whether the “n-altTS vs. in-altTS” difference or “armR vs. armL” difference is larger in specific regions, the difference directly between groups is tested with the feature information such as signal change or the correlation intensity. To avoid confusing, we has rephrased the sentence and added the associated statistical analysis outcome in Page 13 Line11 as “whereas there was no significant difference in the average statistical value of both arm sides; average F-value for armL and armR were 6.06 and 6.12 respectively, P = 0.84 after two-sample t-test.”.
Page 15, “In Table 2, the main arm effect showed a significant impact of thermal perception on the affected arm side (both regions, p-value < 0.05), and the thermal main effect showed a more significant impact on the applied temperature (both regions, p-value < 0.01).” Please rephrase “the thermal main effect showed a more significant impact on the applied temperature” because applied temperature is the independent variable so it will not be affected by the thermal main effect.
=> Thank you for the suggestion. We have rephrased the sentence as follows. “……. the thermal main effect showed a more significant impact on the brain responses by the applied temperature (both regions, p-value < 0.01).”
Page 19, Line 3, Coghill et al., 1999 does not support the statement “the sensory-discriminative processing 2 of pain was confined to the SMA”.
=> Thank you for your suggestion. As suggested, the reference was deleted.
In discussion section, it should be stated clearly that the observed alteration is based on comparison with innocuous TS, instead of the effect of thermal treatment which may be taken as post-altTS vs. pre-altTS.
=> Thank you for your comment. In this study, the observed statistical maps demonstrated the comparison outcome of functional alteration (post-altTS) for the manipulated factors. Within each statistical testing, the pre-altTS data was used as the controlled covariate variable for the correction of baseline heterogeneity. The detail description was described in Page 10 as “The pre-altTS data as the covariate variable was applied to decrease the variation in the regression model caused by heterogeneity of baseline between compared independent variables.”. We revised the discussion section to make more clearly to the readers that the observed alteration is based on comparison with innocuous TS as follows.
Page 18, “The main results supported our hypotheses and indicated that n-altTS induced a prominent functional alteration of motor-related activity that is dependent on the arm side affected, in comparison to innocuous altTS.”
Page 19, “These findings supported the notion that ……. functional plasticity is also dependent on the arm side affected.” Changed to “These findings supported the notion that ………functional plasticity is also dependent on thermal temperature and the arm side affected.”
Page 20, “In this study, we have focused mainly on examining the thermal effect on cortical excitability and have overlooked the fact that functional alteration is affected by the arm side treated.” changed to “In this study, we have focused mainly on examining the effect of different thermal temperatures on cortical excitability and have overlooked the fact that functional alteration is affected by the arm side treated.”
This study reported an arm side difference of the thermal effect. Is there previous study showing this difference in terms of treatment outcomes?
=> Thank you for your comment. The promotion of motor recovery assessed by clinical outcome measures was observable in patients with stroke. However, as we known, no research is done to analyze the arm side difference of the altTS based on clinical measures.
Typo:
Page 4, Line 4: “cannot hardly be determined” should be “can hardly be determined”
=> Thank you for the suggestion. “cannot hardly be determined” is changed to “can hardly be determined”.
Figure 4 caption “The thermal effect for armR shown in (a) the right precentral cortex and in (b) the right postcentral cortex. The effect for armL shown in (c) the left precentral cortex and in (d) left postcentral cortex.” “armR” and “armL” are flipped.
=> Thank you for your comment. Corrections are made in Figure caption.
Page 20, Line 2, “different” should be “differently”
=> Thank you for the suggestion. We changed the word “different” to “differently”.

Round 2
Reviewer 2 Report
The revised manuscript addressed most of my comments. One minor change is needed. Page 13 (Line 7-9), please remove the t-test for F values as comparing F values will not tell which group has more changes.
Author Response
Thank you for the comment. The point-by-point response to is attached.
